# Extended Spectrum β-Lactamase Producing Lactose Fermenting Bacteria Colonizing Children with Human Immunodeficiency Virus, Sickle Cell Disease and Diabetes Mellitus in Mwanza City, Tanzania: A Cross-Sectional Study

**DOI:** 10.3390/tropicalmed7080144

**Published:** 2022-07-22

**Authors:** Maria M. Said, Delfina R. Msanga, Conjester I. Mtemisika, Vitus Silago, Mariam M. Mirambo, Stephen E. Mshana

**Affiliations:** 1Department of Clinical Laboratory, Kondoa Town Hospital, Kondoa P.O. Box 40, Tanzania; mariahsaidy@gmail.com; 2Department of Microbiology and Immunology, Weill Bugando School of Medicine, Catholic University of Health and Allied Sciences, Bugando, Mwanza P.O. Box 1464, Tanzania; conjestermtemisika@yahoo.com (C.I.M.); silago.silago2@gmail.com (V.S.); mmmirambo@gmail.com (M.M.M.); stephen72mshana@gmail.com (S.E.M.); 3Department of Paediatrics and Child Health, Weill Bugando School of Medicine, Catholic University of Health and Allied Sciences, Bugando, Mwanza P.O. Box 1464, Tanzania

**Keywords:** colonization, children, extended spectrum beta-lactamase, *Escherichia coli*, *Klebsiella pneumoniae* complex

## Abstract

Rectal carriage of extended spectrum β-lactamase-lactose fermenters (ESBL-LF) is the major risk factor for the development of subsequent endogenous infections. This study determined the patterns and factors associated with the rectal carriage of ESBL-LF among children with Human Immunodeficiency Virus (HIV), Diabetes Mellitus (DM), and Sickle Cell Disease (SCD) attending clinics at different health care facilities in the city of Mwanza, Tanzania. A cross-sectional study was conducted among children living with HIV (*n* = 236), DM (*n* = 42) and SCD (*n* = 126) between July and September 2021. Socio-demographic and clinical data were collected using a structured questionnaire. Rectal swabs/stool samples were collected and processed to detect the rectal carriage of ESBL-LF following laboratory standard operating procedures (SOPs). Descriptive statistical analysis was conducted using STATA 13.0. The overall prevalence of ESBL-LF carriage was 94/404 (23.3%). Significantly higher resistance was observed to ampicillin, trimethoprim-sulfamethoxazole, and tetracycline among Enterobacteriaceae isolated from HIV infected children than in non-HIV infected children (*p* < 0.05). The commonest ESBL allele 45/62 (72.6%) detected was *bla*_CTX-M_. Generally, a parent’s low education level was found to be associated with ESBL-LF colonization among children living with HIV; (OR 4.60 [95%CI] [1.04–20], *p* = 0.044). A higher proportion of ESBL-LF from DM 10/10 (100%) carried ESBL genes than ESBL-LF from HIV 37/56 (66.1%) and SCD 15/28 (53.6%), *p* = 0.02. There is a need to collect more data regarding trimethoprim-sulfamethoxazole (SXT) prophylaxis and antibiotic resistance to guide the decision of providing SXT prophylaxis in HIV-infected children especially at this time, when testing and treatment is carried out.

## 1. Introduction

Extended Spectrum Beta-Lactamase (ESBL) are enzymes produced by Gram-negative bacteria mainly among members of the family Enterobacteriaceae. Infections associated with ESBLs-producing bacteria are linked to increased costs, morbidity, and mortality [1,2]. Rectal carriage of ESBL-producing bacteria is a potential endogenous source of infections and an independent risk of developing subsequent infections due to ESBL-producing bacteria [3,4,5]. Furthermore, the colonization of ESBL-producing bacteria was found to be high in individuals with comorbidities such as Human Immunodeficiency Virus (HIV) [6], diabetes mellitus (DM), organ transplant, and sickle cell disease (SCD) due to the fact that these patients are often in long-term antibiotic prophylaxis [7,8]. 

The burden of ESBL colonization varies in different populations and geographical locations due to the presence of different drivers of the emergence and spreading of ESBL-LF [9]. Data shows that the burden of ESBL-producing bacteria intestinal carriage is low in high-income countries (HIC) compared to low- and middle-income countries (LMIC) [10,11,12,13,14,15,16,17]. In Tanzania, the prevalence of ESBL rectal colonization has been found to range from 16.5% among adults and children in Mwanza to 54.6% among neonates at Bugando medical Centre [18,19,20,21,22]. In addition, a study in Dar es Salaam reported the ESBL-producing bacteria rectal colonization among HIV children to be 89.7% compared to 16.9% among non-HIV infected children [15]. ESBL families CTX-M, TEM, and SHV are among the commonest enzymes conferring resistance towards β-lactams at our setting [23].

Given these considerations and the lack of information globally, this study determined the patterns and associated factors of ESBL-LF colonization in children with SCD, HIV, DM attending clinics at Bugando Medical Centre (BMC), Sekou Touré regional referral hospital (SRRH), and Baylor Children Hospital (BCH) in Mwanza, Tanzania.

## 2. Materials and Methods

### 2.1. Study Design, Duration, Population and Setting

This hospital-based cross-sectional study was conducted from July to September 2021. The study included children living with HIV infection, diabetes mellitus (DM), and sickle cell disease (SCD), aged from 0–13 years and attending the pediatric outpatient clinic at Bugando Medical Centre (BMC), Sekou Toure Regional Referral Hospital (SRRH) and Baylor Children hospital (BCH). BMC is a zonal consultant and teaching hospital serving the Lake zone regions with a population of about 16 million. BCH is a zonal hospital providing treatment and care for children living with HIV infection and tuberculosis (TB), and SRRH is serving a population of about 2.773 million from Mwanza region.

### 2.2. Data and Sample Collections

Data were collected using a questionnaire customized in Epicollect5 (https://five.epicollect.net/, accessed on 30 September 2021) on smartphone by trained research assistants. Social demographic (e.g., age, gender and place of residence), clinical information (e.g., history of previous hospital admissions and antibiotics exposure), and other relevant information related to risk for ESBL colonization were collected. Rectal swabs were obtained by inserting a single time sterile swab (Guangzhou, China PC: 510530) approximately 2.5 cm and rotated twice in the rectal canal. The swab was removed from the rectal canal after ten seconds, inserted into Stuart transport medium (Guangzhou, China PC:510530), and transported to the Catholic University of Health and Allied Sciences (CUHAS) microbiology Laboratory. In case of rectal swab inconvenience, the participant was provided with a stool container and asked to collect a scoop of feces as an alternative.

### 2.3. Microbiological Procedure

Collected rectal swabs/stool were directly inoculated on two media: plain MacConkey agar (MCA) for isolation of Gram-negative lactose fermenting bacteria, and MacConkey agar supplemented with 2 µg/mL of cefotaxime (MCA-C) for selecting potential extended spectrum β-lactamases producing Enterobacteriaceae (ESBL-PE). All plates were incubated at 37 °C for 18–24 h [24]. A single colony was picked from predominant lactose fermenting colonies with similar morphology for purity plate and further identification using in-house biochemical tests [25]. Isolates were tested for their antibiotic susceptibility patterns using disc diffusion methods by Kirby–Bauer technique [26], and zones of inhibitions were interpreted as per Clinical Laboratory Standard Institute (CLSI) guidelines 2021 [27,28]. Antibiotics that were disk tested included: ampicillin (AMP 10 µg), trimethoprim-sulfamethoxazole (SXT 25 μg), tetracycline (TET 30 μg), gentamycin (CN 10 μg), ciprofloxacin (CIP 5 μg), cefotaxime (CTX 30 μg), piperacillin-tazobactam (TZP 110 μg), amikacin (AK 30 μg), ceftriaxone (CRO 30 μg), Ceftazidime-clavulanic acid (CAL 30 μg), Ceftazidime (CAZ 30 μg), Cefepime (FEP 30 μg), and Meropenem (MEM 30 μg), all from Oxoid, UK. ESBL production was confirmed using the disc combination method. A disc which contains third generation cephalosporins alone (cefotaxime, ceftazidime) and another disc with a combination of clavulanic acid and third generation cephalosporins were seeded on Muller Hinton Agar. The test was considered positive if the difference of inhibition zone diameters between disc of 3rd generation in combination with clavulanic acid and 3rd generation cephalosporin alone was ≥5 mm [28].

### 2.4. Molecular Confirmation of ESBL Enzymes Production

All phenotypic confirmed ESBL-PE were tested for the presence of *bla*_CTX-M_, *bla*_SHV_, and *bla*_TEM_. Bacterial DNA was extracted using the heat treatment method as previously described [29] with some minor modifications. Briefly, loop full colonies of overnight growth of bacteria were suspended into Eppendorf tube containing 500 µL DNase/RNase free water, mixed by vortexing and boiled at 100 °C for 10 min. Tubes were centrifuged at 1200 rpm for 10 min, and 100 µL of the supernatant (DNA rich) was placed in an Eppendorf tube and stored at −20 °C before being used for the detection of *bla*_CTX-M_, *bla*_SHV_, and *bla*_TEM_. Quantity of extracted DNA was determined based on the fluorescence intensity of fluorescent dye binding to double-stranded DNA (dsDNA) using Qubit (Thermo Scientific, 33 Marsiling Industrial Estate Road 3, Singapore ), and quality of extracted DNA was evaluated by electrophoresis in 1.5% agarose gel.

### 2.5. Multiplex PCR Amplifications 

A total of 4 µL of DNA samples was used in multiplex polymerase chain reaction (PCR) to detect ESBL genes. This was performed using a specific set of primers targeting ESBL resistance genes (*bla*_CTX-M_, *bla*_SHV_ and *bla*_TEM_), as shown in Table 1 [30]. The *bla*_CTX-M_, *bla*_TEM_, and *bla*_SHV_ were selected for amplification because these alleles are the commonest genes encoding for ESBL production in the study setting. Multiplex PCR was carried out in a final volume of 25 µL reaction containing HotStarTaq^®^ DNA polymerase mastermix (QIAGEN, Qiagen Str. 1, 40724 Hilden, Germany) and each primer at a reaction concentration of 10 mM. PCR reactions were carried out in thermal cycler machine (BIO-RAD, T100™, WA, USA) and consisted of three steps: initial denaturation at 95 °C for 15 min; 30 cycles of denaturation at 94 °C for 30 s, annealing at 60 °C for 30 s, and extension at 72 °C for 2 min; and a final extension at 72 °C for 10 min.. The amplified products were visualized by electrophoresis in 1.5% agarose gel in TBE (Tris Borate-EDTA) and stained with SYBR green dye safe (Thermo Scientific, 33 Marsiling Industrial Estate Road 3, Singapore). A 100 bp DNA ladder (New England Biolabs; MA, USA) was used as a DNA marker. 

### 2.6. Quality Control

*E. coli* ATCC 25,922 and *E. coli* ATCC 35,218 were used as control strains to optimize the performance of the culture media and antibiotic discs. 

### 2.7. Data Management and Data Analysis

Each participant was given a unique identification number. Laboratory data were recorded in the counter book and then transferred to an Excel sheet tallying with specific demographic and clinical data for cleaning and coding. Data analysis was done using STATA software version 15 (Texas, USA). The categorical variables; sex, residence, parent economic status, education level etc., were presented as proportions, whereas the continuous variables such as age, and number of relative living in the same house were summarized as median (IQR: interquartile range). A Pearson Chi square test was used to compare ESBL proportion in various groups (i.e., DM, HIV and SCD). Stepwise logistic regression model was used to determine factors associated with the colonization of ESBL-LF among children with chronic diseases. Factors with *p*-value less than 0.05 in univariable analysis were subjected to multivariable analysis and factors with *p*-value < 0.05 were considered statistically significant.

## 3. Results

### 3.1. Social-Demographic and Clinical Characteristic of Enrolled Children

A total of 404 children with a median age of 5 (interquartile range [IQR]): (1–13) years were enrolled. More than half (217 or 53.7%) of the participants were males. Almost two thirds (64.6% or 261/404) of the participants’ parents/guardians had a primary education and about three quarters (71.5% or 289/404) were unemployed. More than three quarters (87.1% or 352/404) of the enrolled children resided in urban areas with a median [IQR] family size of 5 (4–7) family members. About 13.8% (40/289) of children and 14.3% (54/377) of their relatives/households had a history of hospitalization in the past three months. Of the 404 enrolled children, more than a quarter (109 or 35.4%) had a history of fever, and 85 (21%) were on antibiotics at the time of enrollment. More than half (236 or 58.4%) were HIV positive and three quarters (*n* = 236) or 75.8% of HIV infected children were on trimethoprim/sulfamethoxazole prophylaxis. About 43% (34.1) children with SCD were on pen V prophylaxis. Moreover, 42 (10.4%) children had DM, and none were on antibiotic use (Table 2).

### 3.2. Bacteria Culture Results 

Out of 404 children, 101 (25.0%) were found to be colonized with isolates resistant to third generation cephalosporins. The predominant isolates were *E. coli* (57.4% or 58/101) followed by *K. pneumoniae* complex (17.00% or 16/101). Using the disc combination method, about 93.1% (94/101) were phenotypically confirmed as ESBL-LF. The prevalence of ESBL carriage among children with chronic disease were 23.8%, 23.7%, and 22.2% for DM, HIV, and SCD, respectively (*p* = 0.945). 

### 3.3. Antibiotics Susceptibility Patterns

Generally, LF Enterobacteriaceae from HIV, SCD, and DM exhibited high resistance to AMP (87.6%), SXT (80.0%), and TE (68.5%). LF isolates from HIV significantly exhibited higher resistance to AMP (90.3%) with *p* = 0.003, SXT (85.1%) had *p* = 0.045, TET (73.1%) *p* = 0.005, AK (16.5%) *p* = 0.027, CAZ (28.3%) *p* = 0.016, and CRO (30.2%) had *p* = 0.012 (Table 3). 

### 3.4. Molecular Characterization of Phenotypically Confirmed ESBL (N = 93)

Among the 94 phenotypic confirmed ESBL-PE, 62 (65.9%) carried at least one gene encoding for ESBL production. The most frequently detected gene was *bla*_CTX-M_, (72.6% or45/62). A combination of two or three ESBL genes carriage was found in 9 (14.5%), and the frequently detected combination was *bla*_CTXM_/*bla*_SHV_ (6 or 9.7%). (Figure 1 and Table 4). Comparing the presence of the ESBL gene, a higher proportion of ESBL-LF from DM 10/10(100%) carried ESBL genes than ESBL-LF from HIV (37/56 or 66.1%) and SCD (15/28 or 53.6%%), *p* = 0.02. 

### 3.5. Factors Associated with ESBL Carriage

Age, sex, residence, parent/guardian level of education and occupation, child history of hospital admission in the past 3 months, child on antibiotic during enrollment, child on antibiotic in the past 3 months, number of members in the household, child on SXT prophylaxis (HIV seropositive only), and child on folic acid/pen V prophylaxis (SCD child only) were included in the model to determine their association with positive ESBL-LF carriage. 

The univariate logistic regression analysis results showed that children whose parent were unemployed (OR 6.0, 95%CI: 1.4–26.1 *p* = 0.004), and being on SXT prophylaxis (OR 2.6, 95%CI 1.1–6.3, *p* = 0.020) were significantly associated with ESBL-LF colonization among children with HIV infection (Table 4). The results from the multivariate analysis showed that parents’ low education level was associated with ESBL colonization (OR [95%CI], *p* value; 4.60 [1.04–20] *p* = 0.044) 

The univariate logistic regression analysis results demonstrated that children with a relative history of admission (OR 2.7, 95% CI: 1.0–7.9, *p*= 0.037) and parents’ education (4.5 [1.69–12.2], *p* = 0.002) were significantly associated with ESBL colonization among children with SCD. Furthermore, according to the results of the multivariate analysis, a lower primary education level of the parent was found to be associated with ESBL colonization (OR [95%CI]; *p* value; 5 [1.54–11.3], *p* = 0.006), (Table 5).

Parent occupation (*p* = 0.002) and history of hospitalization (*p* = 0.001) were statistically significant associated with ESBL carriage among children with DM (Table 6). 

## 4. Discussion

This study, conducted for the first time in Mwanza, represents the prevalence of ESBL carriage among children with chronic disease as 23.8%, 23.7%, and 22.2% for DM, HIV, and SCD, respectively, with no statistically significant differences despite the fact that children with HIV and SCD are on daily use of trimethoprim/sulfamethoxazole and Pen V, respectively. However, the study observed that Enterobacteriaceae isolates from HIV-infected children were significantly more resistant to ampicillin, trimethoprim/sulfamethoxazole, and tetracycline. The observed prevalence was significantly high compared to that which was found in the study done on the general population in the same setting, which observed a prevalence of 16.3% [22] and significantly low compared to the study conducted in the same setting among street children, which found a prevalence of 31.8% [19], signifying the role of hygiene in the transmission of ESBL-LF and variations among different population. The high prevalence in children with co-morbidities could be explained by the fact that children with chronic diseases are at increased chances of using antibiotics and being admitted to hospital. Previous studies documented that hospitalization and prior antibiotic use were the risk factors for ESBL colonization [31].

Nearly one third of the phenotypic confirmed ESBL-LF were not typed by multiplex PCR compared to a study done in the same setting [23] that typed 93.3%; this could be explained by the fact that the primers used in the current study targeted the three ESBL genes *bla*_SHV_, *bla*_TEM_, and *bla*_CTX-M_, and untyped isolates may be harboring other ESBL genes like *bla*_OXA_, *bla*_PER_, *bla*_VEB_, and *bla*_BEL_, which have been documented in our setting [32]. The *bla*_CTXM_ has been the most frequently detected ESBL allele among ESBL-LF in our setting and other settings [23,33]. The predominance of the *bla*_CTXM_ gene may be due to the fact that these genes are encoded by conjugative epidemic plasmids, i.e., IncFII, which play an important role in the effective spreading of this allele [34]; these plasmids have been commonly detected in this setting [35].

The majority of bacterial isolates identified in this study were resistant to multiple classes of antibiotics, as previously reported [36]. Among the three groups, isolates from children with a HIV infection showed significantly high resistance to ampicillin, trimethoprim/sulfamethoxazole, and tetracycline. Similar findings in the same region reported that HIV-infected children had significantly higher multidrug resistance than non-HIV infected children [6]. The level of resistance reflects the high use of these antibiotics in the healthcare system and the community [37]. It should be noted that children living with a HIV infection usually receive long-term trimethoprim/sulfamethoxazole for prophylaxis, as recommended by WHO guidelines [38], and ampicillins are commonly used as the antibiotic of choice in children presenting with upper respiratory infections in primary health care in the community setting. 

Furthermore, isolated bacteria generally exhibited a lower resistance to meropenem than other antibiotics, as previously observed in this setting and other settings in Africa [24,39]. The low resistance of bacteria towards meropenem can be explained by the unavailability of this antibiotic in low tiers of healthcare facilities, its high cost in community pharmacies, and generally, its prescription in regional and tertiary hospitals is guided by the presence of culture results.

In this study, we observed that children with a HIV infection and those with SCD whose parents have a low education level were significantly more colonized with ESBL-LF. Low parent educational attainment is associated with low socioeconomic status, which has been associated with sub-optimal hygiene, which favors the emergence and spreading of ESBL-PE, as previously reported [19,40]. Furthermore, participants’ or relatives’ history of hospital admission was associated with ESBL colonization in children with DM, as observed previously [41]. Usually, the majority of hospitalized patients are on antibiotics treatment, hence the antibiotic selection pressure that increases the chances of ESBL colonization [8,24]. 

## 5. Conclusions

A significant proportion of children with DM, HIV, and SCD was colonized with ESBL-PE in the city of Mwanza with HIV-infected children being more colonized by Enterobacteriaceae resistant to AMP, SXT, and TET than non-HIV infected children. Moreover, the *bla*_CTX-M_ allele was uniformly present in ESBL isolates from children with DM, HIV, and SCD. Nevertheless, children whose parents had a low level of education were significantly more colonized by ESBL-PE. Clinicians should consider ESBL-PE sepsis in children with chronic diseases and institute appropriate management early to prevent associated morbidity and mortality, because in most cases, colonization precedes invasive infections. There is a need to collect more data regarding SXT prophylaxis and resistance to guide the decisions on the practices of providing SXT prophylaxis in HIV-infected children, especially at this time, when testing and treatment is carried out.

## Figures and Tables

**Figure 1 tropicalmed-07-00144-f001:**
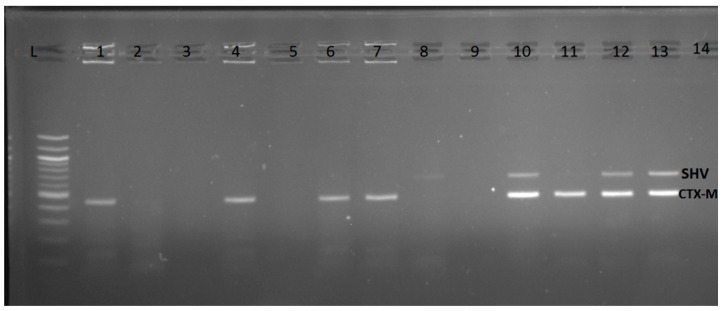
Gel image showing: L = DNA ladder 100 bp; Lanes 1–12 = PCR products; Lane 13 = positive control; and Lane 14 = negative control.

**Table 1 tropicalmed-07-00144-t001:** List of primers used in multiplex PCR.

Gene	Sequence	Amplicon Size	Reference
*bla* _SHV_ f *bla* _SHV_ r	5′-ATGCGTATATTCGCCTGTG-3′ 5′-TGCTTTGTTATTCGGGCCAA-3′	747	[30]
*bla* _TEM_ f *bla* _TEM_ r	5′-TCGCCGATACACTATTCTCAGAATGA-3′ 5′-ACGCTCACCGGCTCCAGATTAT-3′	445	
*bla* _CTX-M_ f *bla*_CTX-M_ r	5′-ATGTGCAGYACCAGTAARGTKATGGC- 3′ 5′-TGGGTRAARTAGTSACCAGAAYCAGCGG- 3′	593	

**Table 2 tropicalmed-07-00144-t002:** Social demographic and clinical characteristics.

Variables		HIV (*N* = 236)	DM (*N* = 42)	SCD (*N* = 126)	Overall (*N* = 404)	*p*-Value *
Median [IQR] age in years (*N* = 404)		4 (2–8)	10 (8–13)	7 (4–10)	6 (3–9)	
Sex (*N* = 404)	Male	122 (51.7%)	18 (42.9%)	77 (61.1%)	217 (53.7%)	0.076
	Female	114 (48.3%)	24 (57.1%)	49 (38.9%)	187 (46.3%)	
Location (*N* = 404)	Urban	209 (88.6%)	28 (66.7%)	115 (91.3%)	352 (87.1%)	0.000
	Rural	27 (11.4%)	14 (33.3%)	11 (8.7%)	52 (12.9%)	
Parent education (*N* = 404)	Primary	175 (74.2%)	26 (61.9%)	60 (64.6%)	261 (64.6)	<0.001
	Secondary	61 (25.8%)	16 (38.1%)	66 (52.4%)	143 (35.4%)	
Parent occupation (*N* = 404)	Employed	35 (14.8%)	26 (61.9%)	54 (42.9%)	115 (28.5%)	
	Unemployed	201 (85.2%)	16 (38.1%)	72 (57.1%)	289 (71.5%)	<0.001
History of fever (*N* = 257)	Yes	32 (27.8%)	6 (37.5%)	53 (42.1%)	91 (35.4%)	0.069
	No	83 (72.2%)	10 (62.5%)	73 (57.9%)	166 (64.6%)	
History of admission past 3-month (*N* = 289)	Yes	3 (2.5%)	7 (16.7%)	30 (23.8%)	40 (13.84%)	<0.001
	No	118 (97.5%)	35 (83.3%)	96 (76.2%)	249 (86.0%)	
Relative history of admission (*N* = 377)	Yes	30 (12.7%)	4 (25.0%)	20 (16.0%)	54 (14.3%)	
	No	206 (87.3%)	12 (75.0%)	105 (84.0%)	323 (85.7%)	0.321
** Current antibiotic (*N* = 404)	Yes	25 (10.5%)	8 (19.1%)	52 (41.3%)	85 (21.1%)	<0.001
	No	211 (89.5%)	34 (80.9%)	74 (58.7%)	319 (78.9%)	
History of antibiotic Past 3-month (*N* = 404)	Yes	44 (18.6%)	9 (21.4%)	56 (44.4%)	109 (27.0%)	<0.001
	No	192 (81.4%)	33 (78.6%)	70 (55.6%)	295 (73.0%)	
Median [IQR] members in household		5 (5–7)	6 (5–7)	5 (4–7)	5 (4–7)	0.001
Type of toilet (*N* = 378)	Modern	199 (84.3%)	14 (87.5%)	119 (94.4%)	332 (87.8)	
	Pit	37 (15.7%)	2 (12.5%)	7 (5.6%)	46 (12.2%)	0.0019
Is the child on Pen V prophylaxis, SCD (*N* = 126)	Yes	NA	NA	43 (34.1%)		
	No	NA	NA	83 (65.9%)		
Is the child on SXT prophylaxis, HIV (*N* = 236)	Yes	57 (24.2%)	NA	NA		
	No	179 (75.8%)	NA	NA		

* two-by-two table was used to calculate *p*-value; ** SXT not included in HIV infected children.

**Table 3 tropicalmed-07-00144-t003:** Antibiotics resistance patterns by isolates source.

DRUG	HIV (*N* = 175) *n* (%)	DM (*N* = 15) *n* (%)	SCD (*N* = 109) *n* (%)	Overall *n* (%)	*p*-Value
SXT	149 (85.1)	10 (66.7)	82 (75.2)	241 (80)	0.045
AMP	158 (90.3)	9 (60.0)	95 (87.2)	262 (87.6)	0.003
TET	128 (73.1)	5 (33.3)	72 (66.1)	205 (68.5)	0.005
CN	40 (22.9)	0 (0.00)	27 (24.8)	67 (22.4)	0.095
AK	21 (12.0)	0 (0.00)	12 (11.0)	33 (11.0)	0.363
CIP	88 (50.3)	5 (33.3)	53 (48.6)	146 (48.8)	0.451
MEM	18 (10.3)	3 (20.0)	8 (7.3)	39 (9.7)	0.240
CAZ	41 (23.4)	4 (26.7)	27 (24.7)	72 (24.0)	0.946
CRO	50 (28.6)	5 (33.3)	32 (29.4)	87 (29.1)	0.924

CIP: Ciprofloxacin, AMP: Ampicillin, MEM: Meropenem, TET: Tetracycline, CAZ: Ceftazidime, SXT: Trimethoprim/Sulfamethoxazole, CN: Gentamicin, CRO: Ceftriaxone, AK: Amikacin.

**Table 4 tropicalmed-07-00144-t004:** The proportions and distributions of ESBL genes among different groups of participants.

ESBL Gene	DM (10), *n* (%)	HIV(56), *n* (%)	SCD(28), *n* (%)	Total(94), *n* (%)
*bla* _CTX-M_	7 (70)	27 (48.2.)	11 (39.3)	45 (47.9)
*bla* _SHV_	0 (0)	1 (1.8)	0 (0)	1 (1.1)
*bla* _TEM_	2 (20)	4 (7.1)	1 (3.6)	7 (7.4)
*bla* _CTX-M/SHV_	0 (0)	4 (7.1)	2 (7.2)	6 (6.4
*bla* _TEM/SHV_	1 (10)	1 (1.8)	1 (3.2)	3 (3.2)
*None*	0 (0.0)	19(33.9)	13 (46.4)	32(34.0)
Total positive	10 (100)	37 (66.1)	15 (53.6)	62 (65.9)

**Table 5 tropicalmed-07-00144-t005:** Factors associated with ESBL-LF colonization among HIV children and SCD.

Variables		ESBL Colonization	Univariate Logistic Analysis		Multivariate Logistic Analysis	
		Positive *n* (%)	OR [95%CI]	*p*-Value	OR [95%CI]	*p*-Value
ESBL colonization among HIV children						
Age		5 [4–6]	1.6 [0.9–1.113]	0.9160		
Sex	Female (114)	22 (19.39)				
	Male (122)	34 (27.87)	1.6 [0.87–2.97)	0.122	1.4 [0.78–2.76]	0.253
Residence	Urban (209)	52 (24.88)	1.9 [0.62–5.7]	0.247		
	Rural (27)	4 (14.81)	1			
Parent/Guardian Education level	Primary (175)	47 (26.86)	2.1 [0.9–4.6]	0.056	4.60 [1.04–20]	0.004
	Secondary (61)	9 (14.75)	1			
Child on SXT	Yes (179)	49 (27.37)	2.6 [1.14–6.30]	0.020	1.7 [0.6–4.67]	0.253
	No (57)	7 (12.28)	1			
ESBL colonization among SCD children						
Age		6.5 [10.5–3.5]	0.97 [0.87–1.09]	0.678		
Parent education level	Secondary (60)	6 (10)	1			
	Primary (66)	22 (33.33)	4.5 [1.69–12.2]	0.002	5 [1.54–11.3]	0.006
Relative history of admission	Yes (20)	8 (40.00)	2.86 [1.0–7.9]	0.037	2.43 [0.8–7.3]	0.116
	No (106)	20 (18.87)	1			
Source of antibiotic	Hospital (21)	2 (9.52)	0.17 [0.33–0.88]	0.024	1.1 [0.45–2.8]	0.768
	Pharmacy (29)	11 (37.93)	1			

**Table 6 tropicalmed-07-00144-t006:** Factors associated with ESBL-LF colonization among DM children.

VARIABLES		ESBL-PE CARRIAGE		*p*-Value *
		POS	NEG	
Age		10 [7–12]	11 [8–13]	0.254
SEX	Female	3 (12.50)	21 (87.50)	
	Male	7 (38.89)	11 (61.11)	0.047
Residence	Urban	1 (28.57)	20 (71.43)	
	Rural	2 (14.29)	12 (85.71)	0.306
Parent education	Primary	2 (12.50)	14 (87.50)	
	Sec	8 (30.77)	18 (69.23)	0.177
Parent occupation	Employed	2 (7.69)	24 (92.31)	
	Unemployed	8 (50.00)	8 (50.00)	0.002
History fever	Yes	3 (50.00)	3 (50.00)	0.424
	No	3 (30.00)	7 (70.00)	
History of Antibiotic	Yes	3 (50.00)	3 (50.50)	
	No	29 (80.56)	7 (19.44)	0.104
Current antibiotic use	Yes	3 (37.50)	5 (62.50)	0.31
	No	7 (20.59)	27 (79.41)	
Past history of admission	Yes	5 (71.43)	2 (28.57)	0.001
	No	5 (14.25)	20 (85.71)	
Relative history of Admission	Yes	2 (50.00)	2 (50.00)	0.551
	No	4 (33.33)	8 (66.67)	
Insulin	Yes	32 (76.19)	10 (23.81)	
Source	Hospital	3 (50.00)	3 (50.00)	
	Pharmacy	0 (0.00)	4 (100.00)	

* two-by-two table was used to calculate *p*-value.

## Data Availability

Not applicable.

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
