# Peer review of "Extended Spectrum β-Lactamase Producing Lactose Fermenting Bacteria Colonizing Children with Human Immunodeficiency Virus, Sickle Cell Disease and Diabetes Mellitus in Mwanza City, Tanzania: A Cross-Sectional Study"

_tropicalmed, 2022, doi:10.3390/tropicalmed7080144_

Round 1

Reviewer 1 Report

Rectal carriage of extended-spectrum β-lactamase-lactose fermenters is a significant risk factor for developing subsequent endogenous infections. The authors investigated the patterns and factors associated with rectal carriage of ESBL-LF among children with HIV, DM, and  SCD attending clinics at different health care facilities in Mwanza, Tanzania, which is very interesting to readers. however, there are some problems to be solved before consideration for publication. (1) How do you judge the inhibition zone diameters more than 5 mm at the antibiotics disk tests, when at least 3 replicates are not consistent? (2)what is the characterization of DNA marker  In gel electrophoresis, readers can not know the size of your DNA marker, and also do not know the size of your amplified fragments. (3) the gel electrophoresis image is not clear and should be replaced.

Reviewer 2 Report

The Manuscript entitled “Extended Spectrum β-lactamase producing lactose fermenting bacteria colonizing children with Human Immunodeficiency Virus, Sickle Cell Disease and Diabetes Mellitus in Mwanza city, Tanzania: a cross-sectional study” is an interesting study presenting the carrier status of ESBL bacteria in children with Human Immunodeficiency Virus, Diabetes Mellitus and Sickle Cell Disease in the city of Mwanza, Tanzania. The manuscript included a large number of patients, is well designed and transmits a strong message about the antimicrobial resistance. I suggest the manuscript be revised by a native English speaker before publication and that the authors carefully check the typos throughout the entire manuscript. 

Specific comments:

1- Please, provide the full name of the abbreviation “STX” in the abstract. 

2-  Line 50: In the sentence “In Tanzania prevalence of ESBL rectal colonization…”, please replace by “In Tanzania, the prevalence of ESBL rectal colonization….”.

3- Line 55: Please, replace the sentence “Given these considerations, and scarce information globally and in the study settings this study determined the patterns…”, by “Given these considerations and scarce information globally, this study determined the patterns…”. 

4- Line 62-68: Please, double check the lack of spaces between the words (e.g. 16million) and the punctuation of the sentence.  

5- Line 84: What is the meaning of ESBL-PE? Please, provide the full name the first time it appears in the text.

6- Line 88: Provide the document of the CLSI in the sentence.

7- Line 93: Exclude the extra “(“ in “(MEM (30 μg)”.

8- Line 115: Please, double check the ordering of the tables description. Here, the correct is Table 1 and not Table 2.

9- Line 110: Exclude “by” from the sentence “….by the good quality of DNA….” and replace “are well separated” by “were well separated”.

10- Table 1: Please, remove “that” from the title “List of primers that used in multiplex PCR”.

11- Line 118: Please, replace “consist” by “consisting”

12- Line 136: Please, replace “live” by “living”.

13- Line 139: Provide the full name of the abbreviation “MDR” as it first appears in the text. 

14- Line 173: Replace “conformed” by “confirmed”. 

15: Line 177: Double check the typos. 

Reviewer 3 Report

This study is of interest to developing countries. It requires making changes to the form, especially the use of acronyms, which is inconsistent.

Acronyms are used when the words are going to be repeated in the text, being necessary to write the complete words with the acronym in parentheses the first time they are mentioned. Then, you must be consistent in its use. the acronym must be explained even though it is in common use and obvious to readers.

For example: ESBL appears and then ESBL-LF and ESBL-PE are used without mentioning why two more letters are added or how they differ from each other. Then, when mentioning the hospital centers, they use acronyms without need, because they are not repeated again in the text. I suggest that at the end of the introduction it is mentioned that the study is carried out in three hospitals in the Mwanza region and then, in the methods, mention them in full, without using acronyms, since they are not repeated in the text again. The same goes for low- and middle-income countries, no acronyms needed.

Introduction: For people who do not know the country, it would be useful to show a map of the area with the location of the centers or, failing that, describe that they are localities around Lake Victoria.

Please specify if these are patients with type 1 or type 2 diabetes mellitus.

Line 113: explain the meaning of PCR.

Line 137: is it Pearson's chi square test?

Line 139: explain the meaning of MDR

Table 2: please use the same format in all tables (no color). P-value: explain at the bottom of the table to which statistical test it corresponds.

Fig. 1: The figure is very simple and does not provide information. It is enough to describe the prevalence it in the text.

Results, line 200: What were the variables included in the model? the adjustment variables used?

Table 5: explain at the bottom of the table the statistical test used

Line 239: microorganisms

Reviewer 4 Report

Extended-spectrum beta-lactamases (ESBLs) confer resistance to most beta-lactam antibiotics, including penicillin and cephalosporin, and the presence of ESBLs-producing organisms is related to poor clinical outcomes. In this manuscript, the authors investigated the factors related to the rectal carriage of ESBL-LF among children with HIV, SCD, and DM in Mwanza city, Tanzania. The authors found that 25% of 404 children carried bacteria resistant to the third-generation cephalosporins. Among these resistant isolates, 93.1% (94/101) were phenotypically confirmed as ESBL-LF. Then, the authors investigated the molecular characterization of phenotypically confirmed ESBL and antibiotics susceptibility patterns. Finally, they analyzed the factors associated with ESBL carriage and found that parent education level, parent occupation, and past admission history were significantly associated with ESBL colonization. Overall, the quality of the data presented should be much more improved. Also, there are many descriptive mistakes causing confusion.  

Comments:

     1.  In section 3.1, some of the percentage calculation is confusing. For example, in line 149, “About 13.8% (n=289) of children and 14.32% (n=377) of their relatives/households had a history of hospitalization in the past three months” how did you calculate the percentage?

      2. In line 156, the results are summarized in table 2 rather than table 1.

      3. It would be better to use bacteria culture results or isolates culture results in 3.2 compared to culture results.

      4. What does Y-axis mean in figure 1?

      5. Could you describe the P-value in table 3? For instance, in the AMP group, the P-value is 0.003. The significance is between HIV-1 and non-HIV-1 patients or other combinations.

      6. The gel imaging should be labelled in figure 2. Also, it would be more helpful to organize section 3.4 results in a table rather than showing a gel picture.

      7. Are there any TEM genes detected in isolates? In table 3, 87.6% of patients are AMP resistant. However, the TEM genes that hydrolyze ampicillin are hard to see in your assay. How do you explain this inconsistency?

Reviewer 5 Report

The authors examine rectal carriage of ESBL in 3 vulnerable groups of children - those living with HIV, Diabetes mellitus and Sickle Cell Disease - in 3 hospital settings in Tanzania and examine socioeconomic risk factors associated with increased carriage rates. The topic is important, given the rising rates of antibiotic-resistance in low and middle income countries and worldwide. The data are noteworthy, however a few points below should be addressed:

1) The Abstract should not use abbreviations without explanation. For example, ESBL-PE is not explained (whereas ESBL-LF is explained), also SXT prophylaxis is not explained so the meanings of both are unclear to the reader.

2) Abstract: The following sentence is confusing and needs to be re-organized: "A significant proportion of children with DM, HIV and SCD 28 were colonized with ESBL-LF with blaCTX-M allele uniformly detected in ESBL isolates from children 29 with DM, HIV and SCD".

3) Materials and Methods section: study design:

The the authors did not note if the study received ethical approval from the relevant ethics committee and if parental informed written consent and/or children's assent was obtained prior to sample collection. Was the study ethically approved and informed written consent and children's assent obtained? This is necessary to ensure that the study is ethically sound.

4)  Materials and Methods section: data collection. The authors mention the use of epicollect software, however was this an online survey or a paper-based survey, was it completed using mobile phones or how was it completed? Did the authors consider literacy issues or access to technology in the completion of this form which is important from a socioeconomic perspective and may have made it difficult for lower socioeconomic individuals to participate?

5) Section 2.3 Microbiological procedure: again ESBL-PE is mentioned but the abbreviation is not explained. As a reader,  I'm unclear of the difference between ESBL-PE and ESBL-LF

6) 2.4 Molecular confirmation of ESBL enzymes production The authors state that 'All phenotypic confirmed ESB-PE were tested for the presence of blaCTX-M, blaSHV and blaTEM'. There is a typo here it should read ESBL-PE. Once again, ESBL-PE should be explained first before being abbreviated. The authors should also state either here or in the introduction that the purpose of evaluating blaCTX-M, blaSHV and blaTEM is because these are the among the most common ESBL encoding genes. This was not described in the introduction and a rationale was not given.

7) Section 2.7 "Stepwise logistic regression model was used to determine factors associated with colonization of MDR among children with chronic diseases." The word MDR should be explained and not abbreviated, especially as it is only used once throughout the manuscript

8) Section 3.1 "The majority of children with SCD 124/125 (98.5%) were on either pen V prophylaxis or Folic Acid (FA) or both." The authors should not combine Penicillin and Folic acid in the total of 124 SCD patients because Folic acid is not an antibiotic. The authors should include the number of patients on Pen V prophylaxis only. Folic acid, a drug which improves red blood cell health in patients with chronic hemolytic disease, is not relevant to a study on resistant antimicrobials and including it here is confusing and irrelevant and it should be removed. In fact, the table shows that 29.4% of SCD children were on 'both' Folic acid and Pen V, while 4.8% were on Pen V alone. Therefore actually only about 35% of SCD children are on prophylactic antibiotics, rather than the 98.5% described in the text. This must be re-worded and excluding Folic Acid entirely makes more sense in the context of antibiotics and antimicrobial resistance. Including it here suggests that the authors are unclear which drug is an antibiotic and which is not. This is a major revision criterion to the paper. Finally, Pen V would be more accurately written as Penicillin V and should be amended accordingly.

The authors also note that all 42 DM children were on insulin therapy, however they should also state if any were on antibiotics (presumably not, but it should be stated regardless).

9) Figure 1, which is just a bar chart of 3 different %s which are clearly outlined in the text is unnecessary, doesn't add any additional information visually and can be removed.

10) Section 3.4 Factors associated with ESBL carriage, the authors describe 'Parent education' as being associated with ESBL. The authors should note whether they mean higher or lower parental education as the risk factor as it is not clear from the text.

11) Discussion section. The authors describe how use of antibiotics e.g. SXT prophylaxis in HIV children can increase risk of ESBL. However the rates of ESBL-LF were the same across all 3 groups of children, including DM children who are not on prophylactic antibiotics. The authors must put forward some suggestion as to why the carriage rates did not differ between the 3 groups, regardless of antibiotic use.

12) Discussion section: "Low education level accounts to low economic status therefore the high ESBL colonization may be accounted by the fact that financial difficulties limit them to access appropriate management of infectious diseases, therefore irrational use of antibiotics". I don't understand why lower education level would result in 'irrational use of antibiotics' as the prescribers of antibiotics are likely to be educated doctors of high socioeconomic background who are not likely to be 'irrational' in their use of antibiotics. Lower socioeconomic status is more likely related to lower access to hygiene e.g. pit toilets and poor hand hygiene which may increase ESBL-LF carriage. The authors should re-frame this sentence.

Round 2

Reviewer 2 Report

Ms. Jana Popovic

Assistant Editor of TropicalMed,

Dear Ms. Popovic,

The manuscript entitled “Extended Spectrum β-lactamase producing lactose fermenting bacteria colonizing children with Human Immunodeficiency Virus, Sickle Cell Disease and Diabetes Mellitus in Mwanza city, Tanzania: a cross-sectional study” was well improved since its first submission and is now suitable to be published in the TropicalMed Journal. All questions raised by this reviewer have been answered. 

Specific comments:

1- Line 184: Replace “conformed” by “confirmed”. 

2- Please double check the font size of the whole text. There are different font sizes throughout the manuscript.

Reviewer 4 Report

The revised version was much improved and can answer my questions, although there were a few writing mistakes. The authors should be more careful.

1.     In Table 2, “is the child on Pen V prophylaxis, SCD (N= 126)” should label 43(34.1%) and 83 (65.9%).

2.     In Table 4, total 10(100%), it is better to remove % to keep consistent with other data.

Reviewer 5 Report

Thank you for addressing many of the comments and improving this manuscript.

I raised this in my initial comments but it has not yet been addressed:

In the discussion section the sentence:

"Low parent education level accounts to low economic status therefore the high ESBL colonization may be accounted by the fact that financial difficulties limit them to access appropriate management of infectious diseases, therefore irrational use of antibiotics."

This does not make sense and must be removed. "Irrational use of antibiotics" implies that the doctors are prescribing antibiotics irrationally or non-sensibly. Why would doctors prescribe antibiotics irrationally to the children of lower economic status individuals?  If lower economic status individuals cannot afford access to healthcare and/or antibiotics, this would limit their antibiotic use and possibly even limit emergence of resistant organisms. This sentence must be removed.

Instead the authors should just include the second explanation they provide. I suggest the following sentence is included only:

"Low parent educational attainment is associated with low socioeconomic status, which has been associated with sub-optimal hygiene, which favors the emergence and spreading of ESBL-PE as previously reported..."

2) If accepted, the manuscript requires a very thorough English language check prior to publication as there are many linguistic and grammatical errors.
